# Biomarkers to Be Used for Decision of Treatment of Hypogonadal Men with or without Insulin Resistance

**DOI:** 10.3390/metabo13060681

**Published:** 2023-05-23

**Authors:** Lello Zolla

**Affiliations:** Dipartimento Scienze Agrarie e Forestali, University of Tuscia, 01100 Viterbo, Italy; zolla@unitus.it; Tel.: +39-392-754-6392

**Keywords:** insulin-resistance, testosterone therapy, hypogonadism, ketone bodies, metabolisms, ketosis, lactate

## Abstract

Male hypogonadism is a result of low testosterone levels, but patients could be insulin-sensitive (IS) or insulin-resistant (IR), showing different impaired metabolic pathways. Thus, testosterone coadministration, which is commonly used to reestablish testosterone levels in hypogonadism, must take into account whether or not insulin is still active. By comparing metabolic cycles recorded in IS and IR plasma before and after testosterone therapy (TRT), it is possible to know what metabolic pathways can be reactivated in the two different groups upon testosterone recovery, and it is possible to understand if antagonism or synergy exists between these two hormones. IS hypogonadism uses glycolysis, while IR hypogonadism activates gluconeogenesis through the degradation of branched-chain amino acids (BCAAs). Upon administration of testosterone, acceptable improvements are observed in IS patients, wherein many metabolic pathways are restored, while in IR patients, a reprogramming of metabolic cycles is observed. However, in both subgroups, lactate and acetyl-CoA increases significantly. In IS patients, lactate is used through the glucose–lactate cycle to produce energy, while in IR patients, both lactate and acetyl-CoA are metabolized into ketone bodies, which are used to produce energy. Thus, in IR patients, an ancestral molecular mechanism is activated to produce energy, mimicking insulin effects. Regarding lipids, in both groups, the utilization of fatty acids for energy (β-oxidation) is blocked, even after TRT; free fatty acids (FFAs) increase in the blood in IS patients, while they are incorporated into triglycerides in those with IR. In both subgroups of hypogonadism, supplementation of useful chemicals is recommended during and after TRT when metabolites are not restored; they are listed in this review.

## 1. Introduction

Male hypogonadism is a disorder characterized by low levels of the hormone testosterone [1]. Besides primary and secondary hypogonadism (the first being related to testicular defects, while the second to hypothalamus or pituitary gland defects), it is known that serum testosterone (T) levels decrease after 50 years of age, affecting 6–12% of men [1,2]. Clinically, testosterone deficiency is associated with metabolic disorders such as hypertension, diabetes, obesity and dyslipidemia [2,3,4], sexual dysfunction and fatigue [5,6]. Biochemically, testosterone plays a significant role in glucose and lipid homoeostasis and their metabolism [7]. Testosterone deficiency, in fact, reduces insulin sensitivity, impairs glucose tolerance, increases fat mass, elevates triglycerides (TGs) and increases low HDL cholesterol. Recent evidence suggests that in hypogonadism, metabolic alterations occurs through different mechanisms [8,9,10,11] depending on whether insulin levels are low or normal [8,12], classifying hypogonadal patients into insulin-resistant (IR) and insulin-sensitive (IS) categories [12,13]. This distinction is made using the HOMAi (Homoeostatic Model Assessment for Insulin Resistance-index). Clearly, inflammatory mediators are different in these two groups, interfering with insulin signaling in different ways. In fact, at low testosterone concentrations, a reduced expression of genes such as solute carrier family 2 member glucose-transporter-type 4 (GLUT4) or (SLC2A4), insulin receptor beta subunit (IR-β), AKT serine/threonine kinase-2 (AKT2), and insulin receptor substrate 1 (IRS-1) was observed, all of which mediate the signaling of insulin, which is responsible for glucose transport. Thus, in hypogonadism showing insulin-sensitivity (IS), over time, insulin levels may increase, leading to insulin resistance (IR) and type 2 diabetes [13,14,15,16,17,18,19,20,21,22,23]. It is well documented that testosterone treatment for hypogonadal men improves insulin signal transduction [24], increasing glucose transporter 4 (GLUT-4) in many tissues, confirming that the GLUT-4 receptor is modulated by testosterone but activated by insulin [25,26]. 

The long-term administration of testosterone is a common testosterone replacement therapy (TRT) in patients with hypogonadism. Treatment can be provided through injectable testosterone esters, transdermal testosterone (gels or patches), or oral testosterone in the form of testosterone undecanoate. All of these delivery modes are acceptable in appropriate doses and allow patients the benefit of having a variety of options to choose from [27]. However, their benefits are still controversial; benefits include better sexual function and bone mineral density as well as increased strength, but cardiovascular risk, especially in older men and younger men with heart disease, has been reported [27]. The final aim of testosterone replacement therapy should not be to increase the serum testosterone level in the medium–normal range nor to resolve the reduction in hypogonadism symptoms, but it is of interest to know which metabolic pathways are or are not restored upon TRT, since testosterone is a hormone influencing metabolomic pathways. Moreover, testosterone replacement should not only be given to men with diagnosed hypogonadism based on persistently low serum testosterone concentrations and symptoms related to low testosterone levels, but it requires measuring all metabolites involved, which will be listed in this review as biomarkers. In particular, it is important to distinguish IR patients from IS patients so as to follow the differences in improvement following testosterone therapy. In fact, as discussed in this review, at the end of TRT, although testosterone values can reach normal values, many metabolomic pathways are not completely restored. This was revealed by comparing all metabolites present in the blood, the final collector of all tissues, before and after TRT. This has been possible by using mass spectrometry (HRMS) methods, which allow us to identify a huge number of metabolites present in human plasma [28,29,30,31]. Metabolomics focuses on the study of low-molecular-weight biochemical molecules (metabolites) in cells, tissues and biofluids. It involves the comprehensive, simultaneous and systematic profiling of many metabolite concentrations and their fluctuations in response to disease, drugs, diet and lifestyle. High-resolution mass spectrometry (HRMS) methods are able to determine a huge number of spectral features in human plasma. Increasing evidence shows that metabolomics could have an impact on the diagnosis, prognosis, drug efficacy and safety of several diseases.

Since testosterone has a complex and different regulatory influence on metabolism in the major tissues involved in insulin action, including liver, adipose tissue and muscle, the analysis of metabolites must to be performed in plasma because it is the final collector of molecules produced from all tissues. Moreover, this approach highlights the importance of taking multiple tissues into account and thus taking a systems biology approach. The comparison before and after testosterone therapy allows one to know the fluctuation of many metabolic cycles in response to testosterone therapy. Finally, since plasma metabolites are determined separately in IS and IR hypogonadism before and after TRT, the cross comparison of metabolites between IS and IR before TRT—where the testosterone is low in both, but insulin is active only in one case—and the subsequent cross comparison of metabolites after testosterone restoration allows us to know if testosterone exerts its action in antagonism or in synergy with insulin. This exhaustive and simultaneous analysis of all metabolites allows for a holistic investigation. The results obtained reveal that in the presence of testosterone, many metabolic cycles are not re-activated either in the presence or in the absence of insulin, and the consequent deficit of some metabolites, different in the two cases, must be supplemented to help patients. This is discussed in this review; the analysis of biomarkers can be helpful to clinicians to evaluate the doses, time of treatment and supplemental chemicals that are to be provided to patients during testosterone treatment.

## 2. Relevant Section

Figure 1 summarizes the main metabolic pathways recorded in the IS and IR plasma from hypogonadism before TRT and after TRT. 

### 2.1. Carbohydrate Metabolism

In IS hypogonadism before TRT treatment, glucose is used as the main biofuel in muscle, adipose and liver, being insulin active. In contrast, in IR, glycolysis is strongly reduced, gluconeogenesis is activated in the liver, and most of the energy comes from the degradation of branched chain amino acids (BCAAs). Thus, it is not surprising that in IR hypogonadism, increased fat and lean body mass are observed, which are absent in IS. On the other hand, the different behavior between IS and IR hypogonadism is expected, since in IR hypogonadism, muscle GLUT4 receptors are reduced [32,33], and glucose is not taken up but instead accumulates in the plasma. In contrast, in the liver, glucose uptake may occur via the GLUT2 transporter, of which gene expression is modulated by testosterone as well as glycogen phosphorylase activity [34,35,36,37]. Moreover, in IR, glucose is produced by gluconeogenesis, as reported by Martin [38].

Upon TRT, glycolysis is significantly increased in both IS and IR, indicating improved glucose utilization related to testosterone restoration, which increases the expression of GLUT4 in cultured skeletal muscle cells, hepatocytes, and adipocytes [38,39], and it promotes glucose uptake [40]. Interestingly, in IR, gluconeogenesis, which is the main energy source before TRT, is stopped after TRT, confirming that the biodegradation of branched amino acids is controlled by testosterone and is a consequence of insulin resistance [41,42]. 

Surprisingly, upon TRT in both IS and IR, lactate and acetyl-CoA increased significantly (Figure 2).

An increase in lactate is the main response to testosterone supplementation, according to Enoki et al. [43] and Burns [44]. The increased lactate production observed in both IS and IR is probably related to reciprocal effects between lactate and testosterone in Leydig cells [14,39,45], where lactate stimulates testosterone production and vice versa. However, lactate production is higher in IR (by about ten times) [30,31], as observed in those with type 2 diabetes and generally in insulin resistance [46], underling insulin’s possible role. Regarding testosterone, recently, it was reported that its deficiency induced a blockage of glucose metabolism in rodent models, favoring the re-programming of metabolic pathways toward glycogen synthesis and a worsening of diabetes mellitus [47]. Interestingly, in IS, the restoration of testosterone increases GLUT4 receptors in muscles, reactivating glycolysis through an anomalous glucose–lactate cycle (Cori cycle) between the liver and the muscles [30], where alanine is excluded from the cycle. Thus, lactate becomes the main energy source for heart, brain, and lungs—the highest oxidative tissues. In these IS patients, utilization of glucose was possible because insulin is active, while in IR, lactate is prevalently metabolized into acetyl-CoA [31], explaining the higher concentration of acetyl-CoA in these patients upon TRT. In fact, a higher lactate concentration induces the feedback inhibition of lactate dehydrogenase, and consequently, pyruvate is converted into acetyl-CoA by PDH enzymes, commonly downregulated by insulin [48]. Furthermore, the degradation of leucine/isoleucine also contributes to an increase in acetyl-CoA in IR [49], causing higher protein catabolism in skeletal proteins and a consequent muscle mass decrease. This is related to the fact that leucine/isoleucine and valine account for nearly 35% of the essential amino acids in muscle proteins, and upon testosterone restoration, higher protein catabolism in skeletal proteins is activated.

Regarding acetyl-CoA increase, it was used differently across the two groups. In IS, it was partly used in the TCA cycle and partly to produce cholesterol (increasing up to 243 mg/dL) [29], while in IR, all the acetyl-CoA was metabolized into the ketone bodies 3-hydroxybutyrate and acetoacetate (Figure 3) [31].

It is of note that ketone bodies are produced only in those with IR [31] and increase with TRT, while their production is inhibited by insulin in those with IS. In fact, 3-hydroxy-3methylglutarylCoa synthase (mtHMGCoA synthase), which catalyzes the breakdown of acetyl-CoA into ketone bodies, is an enzyme inhibited by insulin but overexpressed by testosterone [50]. Consistent with this, treatment of Leydig cells with ethanol blocked testosterone and ketone body production [51].

Thus, ketosis seems to be an alternative pathway for suppling energy in those with IR hypogonadism, mimicking similar metabolic effects of insulin but at a basic control level, bypassing the complex signaling pathway of insulin.

### 2.2. Lipid Metabolism

It is well known that when glucose cannot be used to generate ATP, it is converted into fatty acids (lipogenesis). In the liver and white adipose tissue [51], fatty acids are incorporated into triglycerides, shifting glycerol consumption [52], or they are released into the plasma [53,54]. In this regard, insulin plays a vital role in modulating lipogenesis, promoting glucose uptake and regulating triglyceride catabolism through the inhibition of hormone-sensitive lipase and contributing to insulin resistance [52]. Thus, it is not surprising that a liver belonging to someone with IR hypogonadism is more prone to lipogenesis, where an overexpression of lipoprotein lipase is induced.

Thus, the β-oxidation of short- and medium-chain fatty acids does not represent an energy source in hypogonadism, explaining the increase in fat mass. Consequently, an increased dyslipidemia as well as a mild increase in body mass index (BMI) was observed in those with insulin-sensitive hypogonadism. Interestingly, in IR male hypogonadism, more acetyl-CoA is transformed into mevalonic acid, a precursor to cholesterol, which increased by up to 243 mg/dL [29].

In IS hypogonadism, glycerol 3-phosphate reacted preferentially with fatty acids, producing more phosphatidylcholine (PC) and glycerophospholipids [55].

Finally, acyl-carnitine, which is fundamental for shuttling fatty acids into mitochondria, is not produced from acetyl-CoA in either IS or IR hypogonadism (Figure 4).

Thus, in agreement with Fukami et al. [56], a reduced β-oxidation of fatty acids was recorded [28,29] in both hypogonadisms, indicating that the β-oxidation of short- and medium-chain fatty acids did not represent an energy source in this pathology. Consequently, an increase in fat mass is expected, as is moderately increased dyslipidemia and body mass index (BMI). An accumulation of fatty acids and triglycerides is also responsible for the lower availability of energy for the heart, skeletal muscles, and kidneys [51].

### 2.3. Amino Acids Metabolism

In IR hypogonadism, BCAAs play a main role since they are utilized to produce energy through glycolysis and the TCA cycle. Moreover, since BCAAs are abundantly present in muscle proteins, their higher catabolism causes a decrease in muscle mass. Consistent with this, an increased catabolism of BCAAs was correlated with insulin resistance [41,42], causing a lower body mass index in patients.

In IS hypogonadism, skeletal protein catabolism is reduced, and less BCAAs are released into the plasma [28], which is consistent with the findings of D’Antona [57], who demonstrated the anti-aging role of BCAAs in mitochondrial biogenesis. This supports the role of testosterone in controlling protein synthesis in muscle [58,59] independent of the presence of insulin.

Proline and lysine increase in the blood in both IS and IR hypogonadism. Their accumulation in plasma is indicative of lower bone formation and less collagen synthesis related to testosterone deficiency [60,61], but it is not indicative of the presence or absence of insulin. In support, less proline and lysine were recorded in both IS and IR hypogonadism upon TRT, justifying the osteoporosis observed in any hypogonadism [62,63].

### 2.4. Other Metabolisms

Carnosine is produced from β-alanine derived from uracil and from histidine, but its production seems to require the presence of testosterone [28,29]; in fact, it increases after TRT [30,31]. Its production is higher in IS, suggesting a synergy of insulin and testosterone in this metabolism. Thus, it is not surprising that muscle weakness, fatigue and mental confusion has been reported in hypogonadism, especially in IR hypogonadism patients. This is in line with Penafiel et al. [58] and Varanoske et al. [63], who demonstrated that intramuscular carnosine attenuates fatigue as well as orchiectomy [58].

## 3. Discussion

During the last decade, many studies have been conducted to evaluate the role of testosterone replacement treatment (TRT) in hypogonadal men, but only in terms of putative improvements in clinical symptoms. Thus, increased libido, increased frequency of erections, deeper voice, swelling of the sebaceous glands, increased muscle mass, increased height, bone maturation, etc. were recorded and documented [2,3,4,5,6]. On the contrary, it is of general interest to determine whether all the metabolic pathways are completely restored by TRT, and if so, when this occurs and if it occurs before any clinical symptoms improve. On the other hand, these metabolic pathways are at the base of the aforementioned clinical symptoms. In particular, it remains crucial to distinguish between patients with IS hypogonadism and IR hypogonadism because the presence or the absence of insulin—even when testosterone is restored—might play a role in metabolism. Interestingly, upon the administration of TRT in IR hypogonadal patients for three months, testosterone levels can be restored, but insulin is reduced (decreasing from 17 to 15 µU/mL), according to Kapoor [23]. Since a synergistic and/or antagonistic action between testosterone and insulin exists [24,25], it is not surprising that this partial insulin reduction can limit the total restoration of all metabolic pathways, and thus, it is of interest to know what changes occur upon TRT. In particular, no attention has been paid to determining if metabolites useful to humans are restored after TRT and, in particular, if dangerous metabolites are also produced. If so, it is important to determine how to intervene during and after the therapy in both cases.

In our investigations, by comparing the metabolites in the plasma of IR hypogonadal patients before and after 60 days of TRT, we evaluated which beneficial metabolic effects were a result of the TRT. Our HRMS metabonomic analysis [28,29,30,31] revealed that approximately 20 canonical biochemical pathways were affected, among which 12 pathways were implicated to a significant extent. Data indicated that not all metabolic pathways were restored upon TRT; new metabolic cycles were reprogrammed in both IS and IR patients after testosterone restoration.

In IS hypogonadism before TRT, glycolysis and glutaminolysis produces energy since insulin promotes glutaminolysis, while in IR, the main source of energy is gluconeogenesis through the degradation of BCAAs and the malate–aspartate shuttle [30,31]. Interestingly, upon TRT, both ATP and NADH levels were low [30,31] in both hypogonadisms, and higher levels of lactate and acetyl-CoA were produced, indicating that energy supply in these patients was not provided through canonical pathways; the increased levels of lactate and acetyl-CoA represented the new energy sources.

In IS, an anomalous glucose–lactate cycle is activated, becoming the main energy source in tissues such as the heart, brain, and lungs, highly oxidative tissues. In these tissues, lactate is taken from the blood and metabolized into pyruvate, but alanine is not involved in the cycle [30]. The activation of the glucose–lactate cycle in IS patients is possible since insulin is still active.

In the case of IR, lactate is converted in acetyl-CoA, a reaction inhibited by insulin, and all the acetyl-CoA is preferentially metabolized into the ketone bodies acetoacetate and 3-hydroxybutyrate [31]. The last reaction is catalyzed by MtHMGCoA synthase, which is inhibited by insulin but overexpressed by testosterone [50]. In fact, ketone bodies have never been recorded in IS hypogonadism [28,30], while recently, another group of researchers found ketone bodies in individuals with type 2 diabetes [64]. Thus, ketosis is the main source of energy production in IR, providing the same metabolic effects as insulin but bypassing insulin’s complex signaling pathways. Both insulin and ketones have the same effects on both the metabolites of the first one-third of the citric acid cycle and on mitochondrial redox states, increasing the hydraulic efficiency of the well-perfused working heart [65]. The hydraulic efficiency of the heart is 28% greater with the metabolism of ketone bodies compared to when the heart metabolizes glucose alone. The fundamental reason for this is because there is an inherently higher heat of combustion in b-hydroxybutyrate compared to pyruvate, the mitochondrial substrate which is the end product of glycolysis. Supporting this, lactate and b-hydroxybutyrate were recently indicated as intermediates of energy-metabolism-regulating cellular functions by controlling metabolic, immune, and other body functions [66]. Similarly, ketone bodies are utilized as an energy source by partially replacing glucose in a diabetic human’s heart [67]. A downside to this is that IR hypogonadal patients upon TRT showed symptoms similar to those following a ketogenic diet [68], such as psychiatric problems in particular [69]. This must be taken into account before the administration of TRT to IR hypogonadal patients.

The reported analysis supports the hypothesis that if hypogonadal patients who still have active insulin (IS) are treated with TRT, IS will not worsen and lead to insulin resistance (IR) such that the metabolic pathways related to testosterone and insulin cannot be easily recovered. Clearly, testosterone deficiency, which can occur in individuals who already have insulin resistance due to reasons other than hypogonadism, results in the loss of the ability to reactivate all metabolic pathways. The evidence that some aspects of metabolism cannot be more easily recovered in IR upon TRT should encourage endocrinologists to administrate testosterone therapy before insulin-resistance sets in.

Finally, these comparisons have revealed that by using a systems biology approach, it has been possible to elucidate metabolic pathway changes in hypogonadism [70], allowing one to better understand the mechanism of “metabolic syndrome” related to insulin resistance. Furthermore, the metabolomic changes reported above provide an accurate list of biomarkers that are useful for evaluating the response of tissues to TRT. Figure 5 summarize the plasma metabolites that change significantly in the two hypogonadisms during testosterone deficiency and after testosterone restoration. They can be assumed to be possible biomarkers for checking if part or all metabolic pathways are working properly.

It is of note that by determining these biomarkers in the plasma, one can have an idea of what is going on in different tissues during TRT. For example, an increase or decrease in BBCAs is an indication of protein synthesis in muscles, as an increase or decrease in proline and lysine is related to collagen synthesis and other reactions, as summarized in this review. Finally, in hypogonadal men, tests for sex hormone-binding globulin (SHBG) (free or non-SHBG/bioavailable) are helpful for an accurate diagnosis [71]. It is well known that many men who present with adult-onset testosterone deficiency have a low level of SHBG associated with obesity, insulin resistance and type 2 diabetes. Once that testosterone deficiency is confirmed, the next step is to differentiate between primary and secondary hypogonadism by measuring LH and FSH.

## 4. Conclusions

In conclusion, by comparing metabolic pathways recorded in those with IS and IR hypogonadism before and after testosterone restoration, it can be affirmed that testosterone deficiency (independent of the presence or absence of insulin) causes a decrease in lactate and acetyl-CoA, TCA cycle reduction, blockage of the production of acetyl-Carnitine and consequently of β-oxidation, and blockage of collagen synthesis and carnosine production. Supporting this, after TRT, all metabolic pathways listed above were restored. However, acetyl-carnitine and consequently β-oxidation was never produced in either hypogonadisms, at least in the short term.

Most worrying is that some metabolites are not produced even if testosterone levels are restored.

## 5. Future Directions

This review underlines the importance of using a systems biology approach to elucidate metabolic pathway changes in hypogonadism and to better understand the mechanism of “metabolic syndrome” related to low levels of testosterone and the associated insulin resistance [70]. The new findings will help in selecting patients who will respond to hormone treatment and provides accurate biomarkers for evaluating responses to treatment, eventually leading to better strategies in preventing systemic complications in patients not fit for hormone replacement therapy.

Clinically, testosterone therapy for those with IR hypogonadism should be integrated with the development of gluconeogenesis precursors and supplementation with amino acids, especially leucine, isoleucine and valine. The addition of citrate and other amino acids could help. Carnosine and β-alanine should be supplemented. Clearly, the negative effects of the absence of insulin in those with IR could be better attenuated by the administration of metformin.

Carnitine and/or acetyl-carnitine supplementation is recommended for both sub-groups of patients. In support of this, it has been shown that carnitine addition inhibits the development of cardiovascular disease, ameliorates age-related sexual dysfunction, and reduces levels of free fatty acids [56]. Several studies have emphasized the effect of carnitine as a replacement therapy in the treatment of hypogonadism to improve male re-productive function, making carnitine an appropriate candidate as a therapy to treat symptoms associated with aging [67].

Regarding the ketosis observed in those with IR after TRT, this could be managed by the utilization of the remedies proposed [68] for “keto flu”: increasing sodium supplements with electrolytes, drinking broth (including bone broth and stock cubes) and increasing magnesium, potassium and dietary fat intake (including avocadoes, MCT, olives, butter, nuts and fat bombs), as well as increasing water intake.

Clinically, testosterone therapy for both IS and IR patients should be integrated with chemicals that are not produced metabolically—even after the restoration of testosterone—but are necessary for the functioning of some organs (see Table 1). Acetyl-carnitine supplementation is recommended for both IS and IR patients; it ameliorates age-related sexual dysfunction, inhibits the development of cardiovascular disease, and reduces free fatty acids [56].

In population-based studies, low testosterone is commonly associated with type 2 diabetes and an adverse metabolic profile. SHBG partially mediates the inverse association between testosterone with diabetes; low testosterone is linked to diabetes via a bidirectional relationship with visceral fat, muscle, and bone [72]. Thus, the implementation of lifestyle measures such as weight loss and exercise that, if successful, raise testosterone and provide multiple health benefits, is the best response for an aging, overweight man with type 2 diabetes and subnormal testosterone levels. In fact, testosterone therapy should not be given to such men until the benefits and risks are clarified by adequately powered clinical trials. Finally, nutraceuticals and functional food ingredients, that are beneficial to vascular health, may represent useful compounds that are able to reduce the overall cardiovascular risk induced by dyslipidemia by acting parallel to statins or by acting as adjuvants in the case of treatment failure or in situations where statins cannot be used [73].

## Figures and Tables

**Figure 1 metabolites-13-00681-f001:**
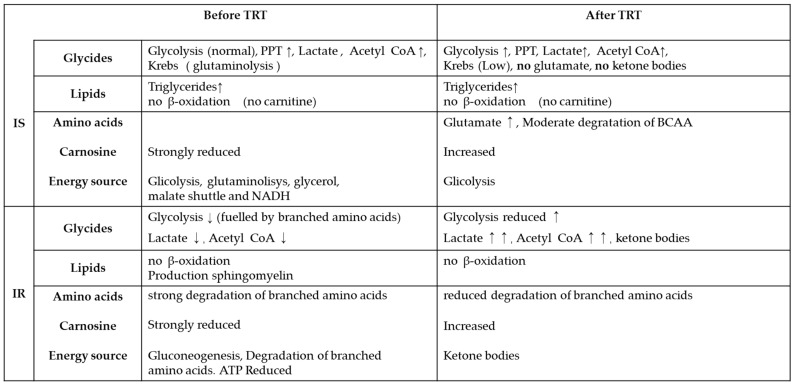
Schematic representation of the metabolisms affected by testosterone deficiency and after TRT in both insulin-sensitive and insulin-resistant hypogonadism. The arrow direction indicates their increased or decreased metabolism and intensity. In the absence of an arrow, their metabolism is under control.

**Figure 2 metabolites-13-00681-f002:**
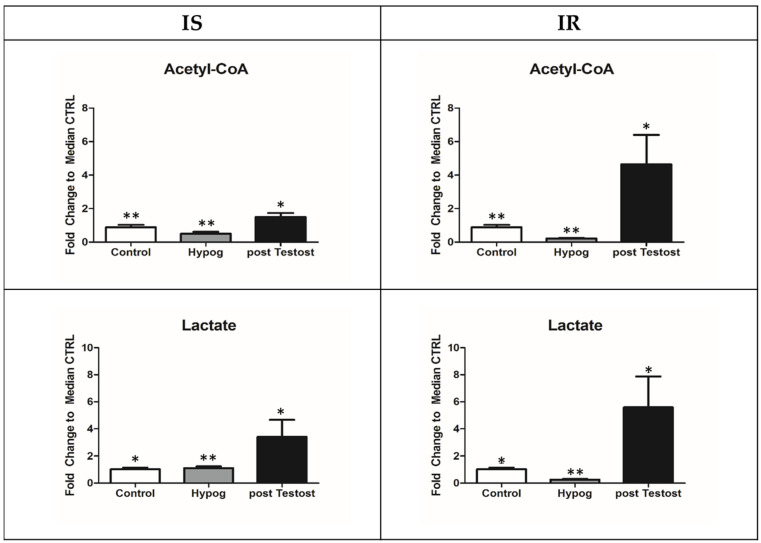
Lactate and acetyl-CoA changes before and after TRT, recorded in the control and hypogonadism subjects. The data showed come from previously published papers [30,31]. Metabolites are expressed as the mean ± SD (*n* = 15) concentration in hypogonadal plasma. * *p* < 0.05, ** *p* < 0.01, against hypogonadal men.

**Figure 3 metabolites-13-00681-f003:**
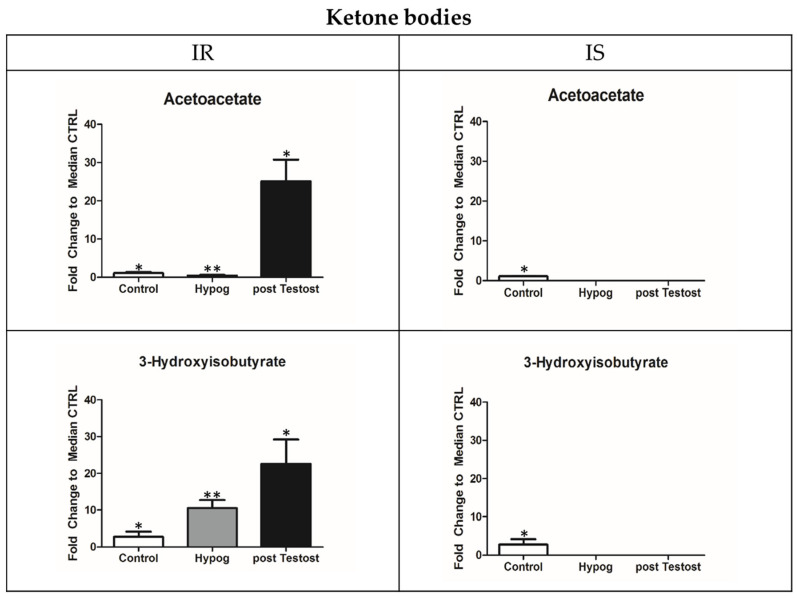
Ketone bodies recorded in the plasma of those with IS and IR hypogonadism before and after testosterone therapy compared to control. The data shown are from previously published papers [30,31]. Metabolites are expressed as the mean ± SD (*n* = 15) concentration in hypogonadal plasma. * *p* < 0.05, ** *p* < 0.01, against hypogonadal men.

**Figure 4 metabolites-13-00681-f004:**
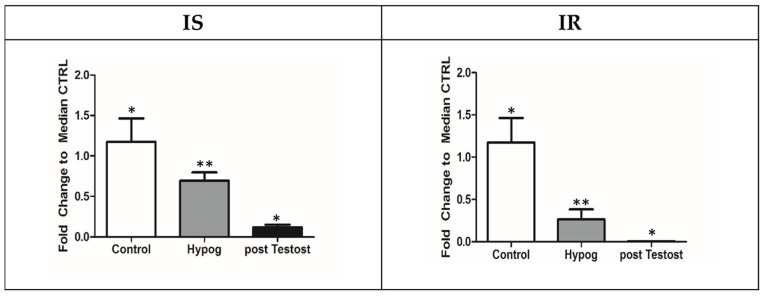
Acetyl-carnitine production during testosterone deficiency and after testosterone restoration. Metabolites are expressed as the mean ± SD (*n* = 15) concentration in hypogonadal plasma. * *p* < 0.05, ** *p* < 0.01, against hypogonadal men.

**Figure 5 metabolites-13-00681-f005:**
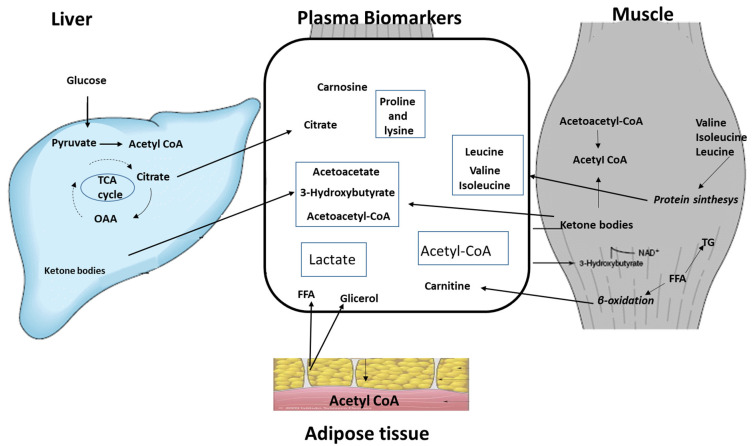
Summary plasma biomarkers that increase or decrease in the plasma of patients with hypogonadism before or after TRT.

**Table 1 metabolites-13-00681-t001:** Chemicals suggested for supplementation during testosterone therapy in two different classes of hypogonadism.

**IS**	Carnosine and β-alanine Lactate
**IR**	Gluconeogenesis precursors Carnosine and β-alanine Carnitine and citrate Amino acids, such as valine and leucine/isoleucine Drinking broth, increasing electrolytes such as potassium and magnesium, dietary fats Increasing water intake.

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
