# Peer review of "Biomarkers to Be Used for Decision of Treatment of Hypogonadal Men with or without Insulin Resistance"

_metabolites, 2023, doi:10.3390/metabo13060681_

Round 1

Reviewer 1 Report

The review describes the differences in response to testosterone treatment of male hypogonadism between subjects with or without insulin resistance. On the base of different biochemical mechanisms staying behind, suitable  biomarkers and  the treatment strategy is suggested.

The collected data are interesting but the paper requires a careful revision as stated below

Major comments

1.  Title. The title is too long and does not express the main  task, which  is identification of the differences in response to testosterone treatment of male hypogonadism between subjects with or without insulin resistance. I may suggest an alternative title  as e.g. “Biomarkers to be used  for decision of treatment of hypogonadal men with or without insulin resistance”.

2.  L.66-69  Moreover, testosterone replacement should not only be given to men with a diagnosis of hypogonadism based on persistently low serum testosterone concentrations and symptoms related  to low testosterone levels, but measuring all metabolites involved..,”

The sentence is too long and the verb is missing. I suggest “Moreover, testosterone replacement should not only be given to men with a diagnosis of hypogonadism based on persistently low serum testosterone concentrations and symptoms related  to low testosterone levels, but it requires measuring all metabolites involved..,”

3.  Table 1.: There are many Italian expressions of biochemical parameters as e.g. “glicidi “ instead of  glycides and many other as “ glicolisis, lattato, glutammate, acetil, triglicerides, osssidation…” which should be written in English.  It occurs also in Fig. 5: “piruvate” instead of pyruvate

4.  Discussion: The authors should mention the relationship between insulin resistance  and sex hormone binding globulin (SHBG) and importance of SHBG as a laboratory marker.. See e.g. Refs.:

Winters SJ. Laboratory Assessment of Testicular Function. 2020 Feb 29. In: Feingold KR, Anawalt B, Blackman MR, Boyce A, Chrousos G, Corpas E, de Herder WW, Dhatariya K, Dungan K, Hofland J, Kalra S, Kaltsas G, Kapoor N, Koch C, Kopp P, Korbonits M, Kovacs CS, Kuohung W, Laferrère B, Levy M, McGee EA, McLachlan R, New M, Purnell J, Sahay R, Singer F, Sperling MA, Stratakis CA, Trence DL, Wilson DP, editors. Endotext [Internet]. South Dartmouth (MA): MDText.com, Inc.; 2000–.

Grossmann M. Low testosterone in men with type 2 diabetes: significance and treatment. J Clin Endocrinol Metab. 2011 Aug;96(8):2341-53.

Author Response

I thank the referee for suggesting a new and more appropriate title for the review. Figure 1 has been corrected and I apologize for the oversights. The bibliography has been added and discussed in the Discussion and Future Directions (in red).

Reviewer 2 Report

This review underlines the importance of using a systems biology approach to elucidate metabolic pathway changes in hypogonadism and for better understanding of the mechanism of “metabolic syndrome” correlated with low levels of testosterone and associated insulin resistance. The new findings will help in selecting patients who will respond to hormone treatment and provide accurate biomarkers for evaluating responses to treatment, eventually leading to better strategies in preventing systemic complications in patients not fit for hormone replacement therapy.

Clinically, testosterone therapy of IR should be integrated with the development of gluconeogenesis precursors as well as supplementation with amino acids, especially leucine, isoleucine and valine. The addition of citrate and other amino acids could help. Carnosine and β-alanine should be supplemented. Clearly, the negative effects of the absence of insulin in IR could be better attenuated by administration of metformin.

Carnitine and/or acetyl-carnitine supplementation is recommended for both subgroups of patients. In support of this, it has been shown that carnitine addition inhibits the development of cardiovascular disease, ameliorates aging-related sexual dysfunction, and reduces levels of free fatty acids. Several studies have emphasized the effect of carnitine as a replacement therapy in the treatment of hypogonadism to improve male reproductive function, making carnitine an appropriate candidate for the therapy of symptoms associated with aging.

Regarding the ketosis observed in IR after TRT, this could be managed by the utilization of remedies proposed  for “keto flu”: increasing sodium supplements with electrolytes, drinking broth (including bone broth and stock cubes) and increasing magnesium, potassium and dietary fat intake (including avocadoes, MCT, olives, butter, nuts and fat bombs), as well as increasing water intake.

Clinically, testosterone therapy of both IS and IR should be integrated with chemicals that are not produced metabolically, even after restoration of testosterone, but necessary for the functioning of some organs.

Author Response

Figure 1 has been improved. Thank you for your suggestion. The data shown has been produced in my laboratory and already published but in a different form and context. Here they are original and require no permissions. The suggested bibliography has been added and discussed

Round 2

Reviewer 1 Report

There are still spelling mistakes in Table 1: glicolysis 'should be glycolysis), glutaminolisis, glicerol... Please check carefuly and use English spelling.

Author Response

Correction of Figure 1 has been done.

Thanks
